# Afil, a Lectin from *Aplysina fistularis*, Exhibits Antibiofilm and Synergistic Antibacterial Activity Against Resistant Bacteria

**DOI:** 10.3390/microorganisms13061349

**Published:** 2025-06-10

**Authors:** Francisco Regivanio Nascimento Andrade, João Marcelo de Sousa Silva, Jéssica de Assis Duarte, Philippe Lima Duarte, Pedro Arthur Sousa Tabosa, Manoel Ferreira da Costa Filho, Juliana Sampaio Nogueira Marques, Alexandre Lopes Andrade, Renata Pinheiro Chaves, Mayron Alves de Vasconcelos, Elielton Nascimento, Ulisses Pinheiro, Edson Holanda Teixeira, Celso Shiniti Nagano, Alexandre Holanda Sampaio, Rômulo Farias Carneiro

**Affiliations:** 1Laboratório de Biotecnologia Marinha—BioMar-Lab, Departamento de Engenharia de Pesca, Universidade Federal do Ceará, Campus do Pici s/n, bloco 871, Fortaleza 60440-970, CE, Brazil; regi.andrade.biotec@gmail.com (F.R.N.A.); joaomarcelo2797@gmail.com (J.M.d.S.S.); jessicaaduarte2@gmail.com (J.d.A.D.); philippe_lima19@hotmail.com (P.L.D.); pedroarthurufc@outlook.com (P.A.S.T.); manoelcostavy@gmail.com (M.F.d.C.F.); renatapinheirochaves@gmail.com (R.P.C.); naganocs@gmail.com (C.S.N.);; 2Laboratório Integrado de Biomoléculas—LIBS, Departamento de Patologia e Medicina Legal, Universidade Federal do Ceará, Monsenhor Furtado, s/n, Fortaleza 60430-160, CE, Brazil; alexandre.andrade@uece.br (A.L.A.); mayronvasconcelos@gmail.com (M.A.d.V.); edsonlec@gmail.com (E.H.T.); 3Faculdade de Educação de Itapipoca (Facedi), Universidade Estadual do Ceará, Av. da Universidade s/n, Itapipoca 62500-000, CE, Brazil; 4Laboratório de Poriferos, Departamento de Zoologia, UFPE—Universidade Federal de Pernambuco, Av. Prof Moraes Rego, 1235, Cidade Universitária, Recife 50670-901, PE, Brazil; biologoefn@gmail.com (E.N.); uspinheiro@hotmail.com (U.P.)

**Keywords:** marine sponge lectin, biofilm inhibition, antibiotic synergy, multidrug-resistant bacteria, lectin–carbohydrate interaction

## Abstract

Lectins from marine sponges have emerged as promising candidates for antimicrobial strategies, particularly against biofilm-forming pathogens. In this study, we report the purification, biochemical characterization, and antibiofilm properties of a new lectin (AfiL) isolated from *Aplysina fistularis*. AfiL exhibited typical features of sponge lectins, including a β-sheet-rich secondary structure and a predominant oligomeric state in solution. Dynamic light scattering (DLS) analyses confirmed that AfiL predominantly exists as a well-defined oligomer at acidic and neutral pH. Sequence analysis revealed similarity to a putative collectin-like protein from sponge *Desydea avara*. AfiL selectively agglutinated *Staphylococcus aureus* strains, correlating with its preferential binding to lipoteichoic acid (LTA). The lectin demonstrated significant antibiofilm activity against *S. aureus*, *S. epidermidis*, and *Escherichia coli* strains, and exhibited synergistic or additive effects when combined with conventional antibiotics against a Methicillin-resistant *S. aureus*. Isothermal titration calorimetry (ITC) revealed a strong interaction between AfiL and porcine stomach mucin (Kd = 1.71 × 10^−6^ M), consistent with multivalent carbohydrate recognition. Overall, our findings highlight the potential of AfiL as a novel antibiofilm agent with species-specific modulatory effects on antibiotic activity and provide new insights into the functional versatility of sponge-derived lectins in microbial control strategies.

## 1. Introduction

Lectins are carbohydrate-binding proteins that play essential roles in key biological processes, including cell recognition, adhesion, and immune responses. Their potential as antibacterial and antibiofilm agents has garnered increasing attention, particularly given the urgent need for alternative therapeutic strategies to combat antibiotic-resistant pathogens [1,2].

The search for new lectins with unique structural and functional properties has driven interest toward marine sponges, which are well known for their exceptional chemical and biological diversity [3,4]. In recent years, several lectins have been isolated and characterized from different sponge species, highlighting their promise as sources of bioactive molecules with antibacterial, antibiofilm, anticancer, and immunomodulatory activities [5,6,7,8,9].

Among marine sponges, the genus *Aplysina*—the most representative of the Aplysinidae family—has yielded four lectins to date: AaL from *Aplysina archeri*, AlL from *A. lacunosa*, ALL from *A. lactuca*, and AFL from *A. fulva*. These lectins have demonstrated relevant biological activities, including bacterial cell agglutination, antibacterial activity, and the inhibition of biofilm formation. Notably, their combination with conventional antibiotics has been shown to enhance antibacterial efficacy, including against resistant strains [8,10,11,12].

Bacterial infections caused by resistant strains represent a significant challenge to public health worldwide. This resistance arises primarily from the adaptive capabilities of microorganisms to conventional antimicrobials, further exacerbated by the overuse of antibiotics and the lack of effective infection control programs. The ability of bacteria to develop resistance, particularly within biofilms, poses a major obstacle to eradication and effective treatment [13,14].

In the present study, we describe the purification and comprehensive biochemical characterization of a new lectin, AfiL, isolated from the marine sponge *Aplysina fistularis*. In addition to structural and thermochemical analyses, we evaluate its antibacterial properties, including its ability to agglutinate bacterial cells, inhibit biofilm formation, interact with bacterial surface components such as lipoteichoic acid (LTA) and lipopolysaccharide (LPS), and enhance the efficacy of antibiotics against multidrug-resistant bacteria.

## 2. Materials and Methods

### 2.1. Animal Collection

Specimens of the marine sponge *Aplysina fistularis* were collected by scuba diving at depths ranging from 18 to 22 m in the Pedra da Risca do Meio Marine State Park. The samples were transported to the laboratory in iceboxes and stored at −20 °C. Species identification was performed by the Department of Zoology at the Federal University of Pernambuco, where voucher specimens were deposited under the code UFPEPOR2712.

All collections were authorized by the relevant environmental agencies (SISBio—Sistema de Autorização e Informação de Biodiversidade, ID: 33913-12).

### 2.2. Purification of the Lectins

Access to the animal’s genetic heritage was granted by the environmental agency SISGEN (Sistema Nacional de Gestão do Patrimônio Genético e do Conhecimento Tradicional Associado, ID: A1792FE).

The lectin purification procedure involved ammonium sulfate precipitation followed by affinity chromatography. Briefly, sponge tissues were cut and macerated in 20 mM Tris-HCl buffer, pH 7.6, containing 150 mM NaCl (TBS), at a 1:3 (*w*/*v*) ratio. The homogenate was filtered through nylon mesh and centrifuged at 9000× *g* for 30 min at 4 °C. The resulting supernatant (crude extract) was precipitated with ammonium sulfate at 70% saturation and maintained at 4 °C for 4 h. After centrifugation under the same conditions, the pellet was resuspended in TBS (fraction F 0–70).

Affinity chromatography was performed using a Sepharose 4B column (1.0 cm × 10.0 cm) equilibrated with TBS. Approximately 25 mL of the F 0–70 fraction was loaded onto the column, which was then washed with TBS to remove unbound proteins. Bound proteins were eluted with TBS containing 300 mM lactose. Fractions showing hemagglutinating activity and absorbance > 0.05 at 280 nm were collected and stored at −20 °C.

Active fractions corresponding to the lectin from *A. fistularis* (AfiL) were pooled, dialyzed, freeze-dried, and stored for further use.

Protein concentrations were determined at each purification step using the Bradford method [15].

### 2.3. Molecular Mass

The molecular mass of AfiL under denaturing conditions was estimated by SDS-PAGE [16], performed in the presence and absence of β-mercaptoethanol (β-ME). SigmaMarker™ Low Range (Sigma-Aldrich, St. Louis, MO, USA) was used as the molecular weight standard. Prior to gel loading, 20 μL of the sample were heated at 100 °C for 5 min.

To assess the molecular mass of AfiL in its native form, freeze-dried protein was solubilized in TBS, centrifuged, and loaded onto a BEH SEC Guard Column (200 Å, 2.5 μm, 4.6 mm × 30 mm) connected to an Acquity UPLC system (Waters Corp., Milford, MA, USA). The column was equilibrated with TBS, and the chromatography was performed at a flow rate of 0.2 mL/min, with absorbance monitored at 280 nm. Calibration was carried out using a gel filtration marker kit (Sigma-Aldrich) for proteins ranging from 29 to 700 kDa.

### 2.4. Hemagglutination Activity

Hemagglutination and inhibition assays were conducted according to established protocols [17] using 3% (*v*/*v*) rabbit erythrocytes (CEUAP—Ethics Committee for the Use of Production Animals ID: 2211202101). Lectin specificity was assessed through hemagglutination inhibition assays [17].

The tested carbohydrates (initial concentration: 100 mM) included the monosaccharides D-xylose, D-ribose, L-fucose, L-arabinose, L-rhamnose, D-galactose, D-mannose, D-glucose, D-glucosamine, D-galactosamine, N-acetyl-D-glucosamine, N-acetyl-D-galactosamine, N-acetyl-D-mannosamine, D-galacturonic acid, and D-fructose. Among the disaccharides and oligosaccharides, the following were tested: D-sucrose, D-melibiose, α-D-lactose, β-D-lactose, D-lactulose, D-maltose, and D-raffinose. Synthetic galactosides and glycoside derivatives included methyl-α-D-galactopyranoside, methyl-β-D-galactopyranoside, methyl-β-D-thiogalactose, phenyl-β-D-galactopyranoside, 4-nitrophenyl-α-D-galactopyranoside, 4-nitrophenyl-β-D-galactopyranoside, and 2-nitrophenyl-β-D-galactopyranoside.

Polysaccharides and glycoproteins tested (initial concentration: 1 mg/mL) included *Saccharomyces cerevisiae* mannan, porcine stomach mucin type II (PSM-II) and type III (PSM-III), fetuin, and bovine submaxillary mucin (BSM).

In addition, the lectin’s interaction with bacterial surface components (initial concentration: 1 mg/mL) was assessed using lipoteichoic acid (LTA) and lipopolysaccharide (LPS).

The effects of pH, temperature, EDTA, and divalent cations on hemagglutinating activity were evaluated according to previously described methods [7]. Briefly, the pH effect was assessed by incubating the lectin in buffers ranging from pH 4.0 to 10.0, while thermal stability was evaluated by exposing the lectin to temperatures between 25 °C and 100 °C prior to the assay. The influence of EDTA and calcium ions was tested by performing the hemagglutination assay in TBS supplemented with either EDTA or CaCl_2_.

### 2.5. Circular Dichroism

The secondary structure of AfiL was analyzed by Circular Dichroism (CD) spectroscopy using a Jasco J-815 spectropolarimeter (Jasco International Co., Tokyo, Japan) equipped with a temperature-controlled Peltier system.

The lectin was solubilized in 20 mM phosphate buffer (pH 7.0) at a final concentration of 0.2 mg/mL. After centrifugation, the sample was transferred to a quartz cuvette with a 1 mm path length. CD spectra were recorded in the far-UV range (190–240 nm) at a scanning speed of 50 nm/min. Secondary structure predictions were performed using the DichroWeb online server [18].

### 2.6. Dynamic Light Scattering (DLS)

DLS analysis was carried out to evaluate the hydrodynamic behavior of AfiL under different pH conditions. The lectin was dissolved at a final concentration of 0.2 mg/mL in three distinct buffers: 20 mM sodium acetate (pH 5), 20 mM sodium phosphate (pH 7), and 20 mM glycine (pH 10). Prior to analysis, all samples were centrifuged at 10,000× *g* for 20 min at 4 °C to remove insoluble aggregates and particulates. The resulting supernatants were transferred to clean, disposable polystyrene cuvettes, and DLS measurements were performed at 20 °C using a Zetasizer Advance Ultra (Malvern Instruments, Worcestershire, UK).

Hydrodynamic diameter distributions were calculated from the intensity autocorrelation functions using the general-purpose analysis model provided by the manufacturer. Each condition was measured in triplicate, and each measurement represents the average of multiple scans. Experimental procedures were adapted from [19,20], with minor modifications to accommodate buffer variation.

### 2.7. Isothermal Titration Calorimetry (ITC)

Isothermal titration calorimetry (ITC) experiments were performed using a MicroCal PEAQ-ITC system (Malvern Instruments). AfiL and the ligands—lactose and porcine stomach mucin (PSM)—were solubilized in TBS at final concentrations of 59 μM (lectin), 5 mM (lactose), and 136 μM (PSM).

Each ligand solution was titrated into the AfiL solution in 2 μL aliquots using an automated microsyringe (Malvern Instruments), with 150 s intervals between injections. A total of 19 injections were made into 250 μL of AfiL solution. During the entire experiment, the sample cell was stirred at 750 rpm and maintained at 25 °C. Control experiments were performed by titrating ligand into buffer alone (TBS) under identical conditions.

Thermodynamic parameters, including dissociation constant (Kd), binding stoichiometry (N), enthalpy change (ΔH), entropy contribution (−TΔS), and Gibbs free energy (ΔG), were calculated using the MicroCal PEAQ-ITC Analysis software v1.41 (Malvern), applying the one-set-of-sites binding model with fitted offset correction, as recommended by the manufacturer. The affinity constant (Ka) was calculated using the reciprocal of the dissociation constant: Ka = 1/Kd.

### 2.8. Determination of the Amino Acid Sequence by Mass Spectrometry (MS/MS)

AfiL was analyzed by SDS-PAGE (12%) as described in Section 2.3. Protein bands were excised from the gel and subjected to reduction and alkylation using dithiothreitol (DTT) and iodoacetamide (IAA), respectively, following the protocol of Shevchenko, et al. [21].

The excised protein was digested with chymotrypsin (Roche, Basel, Switzerland) and trypsin (Promega, Madison, WI, USA), and the resulting peptides were extracted according to the same protocol. Peptide separation was performed on a C18 reverse-phase column connected to a nanoAcquity UPLC system. The column was equilibrated with 0.1% formic acid, and peptides were eluted using a linear gradient of acetonitrile (10% to 85%) containing 0.1% formic acid.

Eluted peptides were infused into a Synapt HDMS hybrid mass spectrometer (Waters Corp., Milford, MA, USA) operating in positive ion mode. The source temperature was maintained at 373 K, with a capillary voltage of 3.5 kV, and data were acquired over an *m/z* range of 100–4000.

For LC-MS/MS analysis, data-dependent acquisition (DDA) was employed, selecting precursor ions with charge states between 2+ and 4+. Collision-induced dissociation (CID) was used for fragmentation, with argon as the collision gas.

Data acquisition and processing were carried out using MassLynx v.4.1 and ProteinLynx v.2.4 software (Waters Corp.). The resulting MS/MS spectra were analyzed manually using the PepSeq tool. Although standard MS/MS does not distinguish leucine from isoleucine due to their identical molecular masses, residue assignment was supported by enzyme-specific cleavage patterns. Since chymotrypsin preferentially cleaves at leucine residues but not at isoleucine, L residues in the sequence were inferred based on observed cleavage sites.

Peptide sequences were compared to known protein sequences using the BLASTp tool available from the National Center for Biotechnology Information (NCBI). Multiple sequence alignments were performed using the MultAlin (http://multalin.toulouse.inra.fr/multalin/ accessed in 12 April 2025) and ESPript 3.0 software packages [22].

### 2.9. Antibacterial Activity

#### 2.9.1. Microorganisms and Growth Conditions

This study employed the bacterial strains *Staphylococcus aureus* ATCC 25923, *S. aureus* ATCC 700698, *S. epidermidis* ATCC 12298, *S. epidermidis* ATCC 35984, and *Escherichia coli* ATCC 11303. *S. aureus* ATCC 700698 and *S. epidermidis* ATCC 35984 are methicillin-resistant strains.

Bacteria were initially cultured on Tryptone Soy Agar (TSA; HIMEDIA, Maharashtra, India) and incubated at 37 °C for 24 h. Isolated colonies were then transferred to Tryptone Soy Broth (TSB) and incubated for an additional 24 h under the same conditions. Bacterial cells were harvested by centrifugation at 9000× *g* for 10 min at 4 °C, resuspended in TSB, and adjusted to a final concentration of 2 × 10^6^ CFU/mL based on optical density at 620 nm.

#### 2.9.2. Bacterial Agglutination Assay

The bacterial agglutination assay was performed using *S. aureus* ATCC 25923 and *E. coli* ATCC 11303. Bacteria were harvested, washed, and suspended in TBS containing 4% formalin, followed by incubation at 4 °C for 24 h. After washing and resuspension in TBS, the bacterial concentration was adjusted to 2 × 10^8^ cells/mL.

AfiL (1 mg/mL) was incubated with the bacterial suspension for 1 h at room temperature. To assess inhibition by carbohydrate, bacteria were pre-incubated with 100 mM lactose for 30 min before lectin addition. Controls included bacterial suspensions with or without lactose. Agglutination was observed under a light microscope.

#### 2.9.3. Effect of Lectin on Planktonic Cell Growth of Microorganisms

The effect of AfiL on planktonic bacterial growth was evaluated by broth microdilution in 96-well polystyrene plates, following Clinical and Laboratory Standards Institute (CLSI M07-A10, 2015 [23]) guidelines, with slight modifications [24].

#### 2.9.4. Lectin Effect on Biofilm Formation

The ability of AfiL to inhibit bacterial biofilm formation was evaluated according to the method described by Vasconcelos et al. [24], with modifications reported by Stepanovic et al. [25]. Biofilms were developed in the presence or absence of AfiL at concentrations ranging from 7.8 to 500 μg/mL for 24 h. Total biomass was assessed using the crystal violet staining method. After biofilm formation, the wells were washed thrice with phosphate-buffered saline (PBS), fixed with 99% methanol, stained with 1% crystal violet, and then washed to remove excess stain. The retained stain was solubilized with 33% acetic acid, and the absorbance was measured at 590 nm using a SpectraMax i3 Multi-Mode Microplate Reader (Molecular Devices LLC, San Jose, CA, USA).

To quantify viable cells within the biofilm, the culture medium was removed, and the wells were washed to eliminate non-adherent cells. PBS was then added to each well, followed by ultrasonic treatment to disrupt the biofilm and release the embedded cells. Serial dilutions were plated on tryptic soy agar (TSA) and incubated at 37 °C for 24 h. Colony-forming units (CFUs) were counted, and results were expressed as Log_10_ CFU/mL.

#### 2.9.5. Combination Activity of Afil and Antibiotics

The effect of AfiL in combination with antibiotics was evaluated using the checkerboard method, as described by [6]. AfiL was tested in combination with ampicillin (Amp) and tetracycline (Tetra) against three bacterial strains. Antibiotics were prepared at ½×, ¼×, 1/8×, and 1⁄16× their respective MICs values, while AfiL was maintained at a fixed concentration of 125 μg/mL. Plates were incubated at 37 °C for 24 h, and bacterial growth was measured by absorbance at 620 nm using a SpectraMax i3 Multi-Mode Microplate Reader.

Additive effect was defined as growth inhibition at ½× MIC, while a synergistic effect was considered when inhibition occurred at concentrations between ¼× and 1⁄16× MIC.

#### 2.9.6. Statistical Analysis

All data were analyzed using GraphPad Prism^®^ version 7.0 for Microsoft Windows^®^. Comparisons were performed using one-way analysis of variance (ANOVA), followed by the Bonferroni post hoc test. A *p*-value < 0.05 was considered statistically significant. All assays were conducted in triplicate in three independent experiments.

## 3. Results

### 3.1. Purification of the Lectin

The aqueous extract from *Aplysina fistularis* exhibited strong hemagglutinating activity against rabbit erythrocytes. The purification of the lectin was achieved through a combination of ammonium sulfate precipitation and affinity chromatography using a Sepharose 4B matrix. The procedure resulted in an 8.4-fold purification, with a yield corresponding to 15% of the total hemagglutinating activity present in the crude extract (Table 1).

### 3.2. Molecular Mass

On SDS-PAGE, AfiL displayed a single band of approximately 70 kDa in the absence of β-mercaptoethanol. Under reducing conditions, two bands of approximately 30 kDa were observed, indicating the presence of disulfide-linked subunits (Figure 1).

Under native conditions, as determined by size-exclusion chromatography, AfiL exhibited a single, symmetrical peak with an estimated molecular mass of 70 kDa, consistent with a dimeric structure stabilized by disulfide bonds.

### 3.3. Hemagglutinanting Assay and Inhibition

The results of the hemagglutination inhibition assay are summarized in Table 2. AfiL was inhibited by D-lactose (Galβ(1→4)αGlc), 4-nitrophenyl-α-D-galactoside, and 4-nitrophenyl-β-D-galactoside, all exhibiting a minimum inhibitory concentration (MIC) of 12.5 mM. The disaccharide D-lactulose (Galβ(1→4)Fru) showed an MIC of 25 mM.

Among the glycoproteins tested, porcine stomach mucin (PSM) was the most potent inhibitor (MIC = 0.006 μg/mL), followed by bovine stomach mucin (BSM) and fetuin, with MICs of 0.049 μg/mL and 0.39 μg/mL, respectively.

In addition to carbohydrates and glycoproteins, AfiL also interacted with key bacterial surface components. LPS, a major component of Gram-negative bacterial outer membranes, inhibited AfiL hemagglutinating activity at an MIC of 250 μg/mL. Similarly, LTA, present in Gram-positive bacterial cell walls, inhibited activity at an MIC of 125 μg/mL. These results support the potential role of AfiL in recognizing conserved bacterial structures involved in pathogenicity and biofilm formation.

AfiL exhibited hemagglutinating activity over a wide pH range, with maximum activity observed between neutral and basic conditions (Appendix A). In terms of thermal stability, the lectin retained its activity up to 50 °C but was completely inactivated at 70 °C (Appendix A).

Furthermore, the presence of calcium ions or the chelating agent EDTA had no effect on hemagglutinating activity, indicating that AfiL is not calcium dependent.

### 3.4. Circular Dichroism

AfiL exhibited a characteristic CD spectrum with a prominent negative band centered at 216 nm, indicating a predominance of β-structures in its native conformation (Appendix A).

Secondary structure content, estimated using the CONTIN algorithm on the DichroWeb online server, revealed that AfiL consists of approximately 58% β-sheet and 42% random coil, with no detectable α-helical content.

### 3.5. Dynamic Light Scattering

DLS revealed distinct hydrodynamic behaviors of AfiL depending on the pH of the buffer (Table 3). At pH 5, AfiL displayed a narrow, monodisperse distribution with a predominant population centered around 29 nm, indicative of a stable and homogeneous oligomeric state. At neutral pH (pH 7), the analysis showed three main size populations by intensity, with peaks at approximately 9, 45, and 220 nm, suggesting partial aggregation or the presence of multiple oligomeric species in equilibrium. At alkaline pH (pH 10), the size distribution became markedly broader, with peaks observed at ~118 nm and 452 nm, and a higher polydispersity index (PDI = 0.60), consistent with increased heterogeneity and the formation of large aggregates.

Across all conditions, distributions by number and volume consistently indicated the presence of smaller species (10–30 nm), whereas distributions by intensity disproportionately emphasized larger particles due to their stronger light-scattering contribution. These results suggest that AfiL is structurally more stable at acidic pH, while at basic pH, it tends to form higher-order aggregates, likely driven by conformational changes or reduced solubility. The full hydrodynamic size distributions by number, volume, and intensity are presented in Appendix A.

### 3.6. ITC

The binding of AfiL to PSM was evaluated by ITC. The resulting thermogram (Figure 2A) and binding isotherm (Figure 2B) were initially fitted using a one-set-of-sites model, revealing a strong exothermic interaction characterized by a dissociation constant (Kd) of 1.68 × 10^−6^ M. The heatmap of integrated heats (insert in Figure 2B) corroborated the strong affinity profile. The enthalpy change (ΔH) reached −80 kcal/mol, corresponding to the lower detection limit of the instrument, suggesting highly enthalpy-driven binding. The stoichiometry (N) was estimated at 0.148, indicating that approximately seven AfiL molecules interact with a single PSM molecule, consistent with multivalent glycan binding typical of highly glycosylated substrates.

The interaction between AfiL and D-lactose was also evaluated by ITC. The thermogram (Figure 2C) and binding curve (Figure 2D) demonstrated a specific interaction fitted to a one-set-of-sites model. The dissociation constant (Kd) was determined to be 1.02 × 10^−5^ M, reflecting a lower affinity compared to PSM. The binding stoichiometry (N) was approximately 2.0, suggesting that each AfiL dimer engages two lactose molecules. The heatmap (insert in Figure 2D) confirmed the specific, although weaker, binding event relative to PSM.

### 3.7. Amino Acid Sequencing

Digestion of AfiL with trypsin resulted in the identification of nine peptides, while chymotrypsin digestion yielded five additional peptides (Appendix A). Together, these peptides represent approximately 42% of the primary structure of AfiL.

Peptide analysis revealed microheterogeneity, with isoforms identified in peptides T5 and T6, and multiple overlapping regions among distinct fragments. For instance, peptide Q1 overlapped with the C-terminal region of peptide T7 and the N-terminal region of peptide T2. Additionally, peptide T2 completely overlapped with peptide Q4. These alignments enabled the reconstruction of a continuous amino acid sequence comprising 32 residues: PPETLDEAYVDGLSLTHGSPRQHLFSYASGWR, with a molecular weight of 3585.71 Da.

The continuous sequence, along with the set of non-overlapping peptides, was subjected to BLASTp analysis. Although no significant homology was found with previously characterized lectins, 50% identity and 59% similarity was observed with a putative protein from the marine sponge *Dysidea avara* (XP_065893106.1). The alignment between AfiL and this putative sequence is shown in Figure 3.

These findings suggest that AfiL represents a structurally distinct lectin and may belong to a previously uncharacterized group of sponge lectins with potential biological relevance.

### 3.8. Antibacterial Activity

#### 3.8.1. Effect on Planktonic Bacterial Growth

AfiL did not inhibit the planktonic growth of *S. aureus* ATCC 25923 or *E. coli* ATCC 11303. However, against the methicillin-resistant strain *S. aureus* ATCC 700698 (MRSA), AfiL exhibited a bacteriostatic effect, with a MIC value of 125 μg/mL.

#### 3.8.2. Bacterial Agglutination Assay

In agglutination assays, AfiL agglutinated *S. aureus* ATCC 25923 but not *E. coli* ATCC 11303. The agglutinating activity was inhibited by pre-incubation with α-D-lactose, indicating a carbohydrate-dependent interaction (Figure 4). These results suggest selective recognition of cell surface glycoconjugates on Gram-positive bacteria.

#### 3.8.3. Inhibition of Biofilm Formation and Viable Cell Counts

AfiL significantly inhibited biofilm formation by *S. aureus* ATCC 25923 at all tested concentrations, despite showing no effect on its planktonic growth. Biofilm biomass was reduced by more than 70%, with consistent inhibition across the concentration range (Figure 5a). Regarding viable cell counts, all concentrations except the lowest (3.9 μg/mL) led to reductions between 0.3 and 1.0 log_10_ CFU/mL (Figure 5f).

For the methicillin-resistant strain *S. aureus* ATCC 700698 (MRSA), AfiL showed a bacteriostatic effect at 125 μg/mL. Biofilm biomass was reduced by over 70% at all tested concentrations, with nearly complete inhibition observed at 62.5 μg/mL (Figure 5b). However, viable cell counts within the biofilm did not decrease; instead, a slight but statistically significant increase (~0.3 log_10_ CFU/mL) was observed at 31.2 μg/mL (Figure 5g).

Similar effects were observed in *S. epidermidis* strains. In ATCC 12228, AfiL reduced biofilm formation by 30–50% across all concentrations (Figure 5c), and significantly reduced viable cell numbers at concentrations from 31.2 to 250 μg/mL, with reductions ranging from 0.2 to 1.0 log_10_ CFU/mL (Figure 5h). In ATCC 35984, a methicillin-resistant strain, AfiL inhibited biofilm biomass by more than 65% at 62.5 and 250 μg/mL (Figure 5d). Viable cell reductions of approximately 0.3 log_10_ CFU/mL were observed at 31.2, 125, and 250 μg/mL (Figure 5i).

For *E. coli* ATCC 11303, AfiL showed the greatest inhibitory effect on biofilm formation among all tested strains, with reductions of 50–75% at concentrations between 15.6 and 250 μg/mL (Figure 5e). Viable cell counts decreased significantly at all concentrations, with the most pronounced reduction (~1.8 log_10_ CFU/mL) observed at 250 μg/mL (Figure 5j).

#### 3.8.4. Synergistic Effects with Antibiotics

The combination of AfiL with ampicillin or tetracycline showed variable effects depending on the bacterial strain (Table 4). For *S. aureus* ATCC 700698, a significant reduction in MIC values was observed. Ampicillin in combination with AfiL exhibited an additive effect, while tetracycline demonstrated a synergistic effect, indicating potential for enhanced therapeutic efficacy.

Conversely, for *E. coli* ATCC 11303, the combination of AfiL with either antibiotic resulted in an increase in MIC values, suggesting an antagonistic interaction in this Gram-negative strain.

These results highlight the strain-specific and context-dependent nature of AfiL’s antibacterial potential, particularly its promising activity against *S. aureus*.

## 4. Discussion

The genus *Aplysina* Nardo (1984) has been extensively studied for its potential in the bioprospecting of natural products, with numerous compounds such as alkaloids and brominated metabolites described in these sponges [26,27,28]. In parallel, *Aplysina* has also emerged as a valuable source of lectins, with four lectins previously isolated from different species [8,10,11], placing it as the second most studied sponge genus in lectinology—second only to *Haliclona*. Still, considering the 15 valid *Aplysina* species reported along the Brazilian coast [29], the number of identified lectins remains limited, highlighting the need for continued bioprospecting in this genus.

As part of our ongoing efforts to expand knowledge of sponge lectins, we investigated several species for hemagglutinating activity and antibacterial activity in aqueous extracts. In this context, we identified AfiL, a new lectin purified from *A. fistularis*, expanding the repertoire of sponge lectins with relevant biological properties. AfiL exhibited characteristic features commonly observed in other sponge lectins, such as optimal hemagglutinating activity at neutral to basic pH, thermal tolerance, cation independence, and a clear specificity for galactosides [30,31,32,33,34].

However, as observed for other *Aplysina* lectins [8,10], free galactose was not sufficient to inhibit hemagglutination, whereas galactose derivatives—such as nitrophenyl-substituted forms—and galactose-containing carbohydrates, including lactose and lactulose, effectively inhibited AfiL activity. Notably, the addition of hydrophobic substituents to the galactose moiety, as seen with nitrophenyl derivatives, significantly enhanced inhibition, while the configuration at the anomeric carbon (α vs. β) had no apparent effect on the MIC values. This behavior contrasts with that of lectins from *Aplysina archeri* and *A. lacunosa*, which show marked preference for β-galactosides [11], and with that of ALL, AcrL, HoL-30 (*Halichondria okadai* lectin), CCL (*Chondrilla caribensis* lectin), CchGs (*Cinachyrella* sp. galectins), and hRTL (*Chondrilla australiensis* lectin) classified as a galectins, which also exhibit β-galactoside specificity [3,4,7,12,35,36].

As observed for some marine sponge lectins [37,38], AfiL displayed high binding affinity to PSM, a glycoprotein densely glycosylated with galactose-rich oligosaccharides. This finding is reminiscent of the strong binding reported for the *Craniella australiensis* lectin [38]. Given the low MIC observed for PSM, we employed ITC to further characterize the interaction, comparing it to lactose, a simple galactoside.

ITC data revealed an association constant (Ka) of 9.80 × 10^4^ M^−1^ for lactose, with a binding stoichiometry of ~2. This is compatible with the expected two carbohydrate recognition domains (CRDs) per AfiL dimer—similar to the stoichiometry observed in lectins from *A. archeri* and *A. lacunosa*, and ALL [11,12], which contains a single CRD per monomer. To our knowledge, these are the first ITC measurements reported for sponge lectins, making direct comparison with previously published Kd values unavailable.

In contrast to the simple monosaccharide, AfiL displayed a significantly higher affinity for PSM (Ka = 5.95 × 10^5^ M^−1^), with a calculated stoichiometry of ~0.14, indicating approximately seven AfiL molecules per mucin. This multivalent interaction likely reflects the presence of numerous glycan motifs within the PSM preparation. Similar behavior has been documented for plant lectins, where certain mucins exhibit hundreds of binding sites, allowing lectins to dynamically transition between adjacent glycans and, thereby, achieve high apparent affinities [39,40].

The structural characterization of AfiL revealed a well-defined oligomeric state in solution. SEC showed a single, symmetrical peak with an estimated molecular mass of ~70 kDa, suggesting a dimeric assembly, a feature commonly observed among sponge lectins [30,37]. This finding was corroborated by DLS, which revealed a dominant population with a hydrodynamic diameter between 10 and 30 nm under acidic and neutral conditions—values compatible with dimers or trimers of globular proteins, indicating a well-defined quaternary structure in solution. Although distributions by intensity indicated the presence of higher-order species, these likely represent minor aggregates.

Complementary MS/MS analyses revealed amino acid sequence similarity between AfiL and a putative uncharacterized protein from *Dysidea avara*, which, based on domain modeling, possesses a collagen-like domain at its N-terminus. While the sequence coverage was limited to the C-terminal, this structural motif is noteworthy because it is also found in collectins—a subfamily of C-type lectins that typically oligomerize through collagen-like domains to perform roles in innate immunity [41]. Although collectins are largely restricted to vertebrates [42], recent genomic studies have identified similar motifs in sponges, supporting a possible ancient origin. In this context, AfiL may represent a structurally conserved form of oligomeric lectin with potential functional parallels, while also contributing to our understanding of lectin evolution in early-diverging metazoans. Further structural and functional analyses are required to determine the true nature and evolutionary classification of AfiL.

AfiL exhibited selectivity in agglutinating *S. aureus*, a Gram-positive bacterium, while showing no agglutination activity toward *E. coli*, a Gram-negative strain. This specificity likely reflects AfiL’s preferential binding to glycoconjugates characteristic of Gram-positive bacterial surfaces, particularly teichoic acids. Supporting this, hemagglutination inhibition assays revealed a lower MIC for LTA (125 μg/mL) compared to LPS (250 μg/mL), indicating affinity for Gram-positive cell wall components. This selective interaction may underlie the bacteriostatic effect observed exclusively against the methicillin-resistant *S. aureus* strain (ATCC 700698), suggesting that surface glycan composition plays a key role in the antibacterial activity of the lectin. Together, these data may reflect a primitive but functionally effective immune-like mechanism in sponges, aiding in the neutralization or immobilization of invading microorganisms [37,43,44].

Beyond agglutination, AfiL’s capacity to significantly reduce biofilm biomass—even in the absence of planktonic growth inhibition—suggests interference with early adhesion processes or extracellular matrix formation, both of which are closely associated with Gram-positive cell envelope architecture. This behavior was particularly evident in *S. aureus* and *S. epidermidis* strains, where AfiL treatment led to substantial reductions in biofilm mass and/or viable cell counts. AfiL likely prevents the establishment or maturation of biofilm by interfering with early adhesion or matrix formation, reducing overall biomass. However, the small subpopulation that remains adherent can still retain viability, which explains the maintained or slightly elevated CFU values in some conditions. This observation is consistent with previous studies on antibiofilm marine lectins and reinforces AfiL’s role in preventing the formation of biofilms rather than bactericidal compound [7,10].

Anti-biofilm effects are particularly significant given the resilience of biofilms in chronic infections and their association with increased antimicrobial resistance. Similar biofilm-inhibitory properties have been reported for other sponge-derived lectins, such as ALL, AFL, and AcrL [7,8,12], reinforcing the therapeutic potential of this protein class.

The observed synergy between AfiL and conventional antibiotics further underscores its potential biomedical applications. AfiL enhanced the efficacy of ampicillin and tetracycline against *S. aureus* ATCC 700698, with a synergistic effect noted for tetracycline. Such results support the potential of sponge-derived lectins as selective antibiotic adjuvants. These proteins may enhance the efficacy of conventional antibiotics in a strain-specific manner, potentially reducing required dosages and delaying the onset of resistance in targeted clinical settings. Similar synergistic effects have been reported with other sponge lectins, including ALL and AcrL [7,12].

Given the urgent global challenge of antimicrobial resistance [13], the discovery of novel antimicrobial lectins such as AfiL offers a promising alternative or complementary path to traditional antibiotics. The integration of marine lectins into combinatory treatment regimens may provide effective and innovative solutions to current therapeutic bottlenecks. Despite the promising in vitro data, this study has limitations that should be ad-dressed in future work. Future studies should aim to validate AfiL’s selectivity and safety profile in vivo, extend its evaluation to additional Gram-negative and biofilm-forming pathogens, and explore its mechanism of biofilm inhibition.

## 5. Conclusions

AfiL, a new lectin from *Aplysina fistularis*, exhibits selective agglutination of *S. aureus*, strong antibiofilm activity, bacteriostatic effects against methicillin-resistant strains, and synergistic interactions with conventional antibiotics. Its specificity for complex galactosides, as well as LPS and LTA, highlights its potential to target key bacterial structures. These findings position AfiL as a promising candidate for the development of alternative antibiofilm-based strategies, particularly for preventing biofilm formation in clinically relevant bacteria. The results also reinforce the biotechnological relevance of sponge-derived lectins as selective modulators of antibiotic activity, enhancing efficacy in a species-specific manner.

## Figures and Tables

**Figure 1 microorganisms-13-01349-f001:**
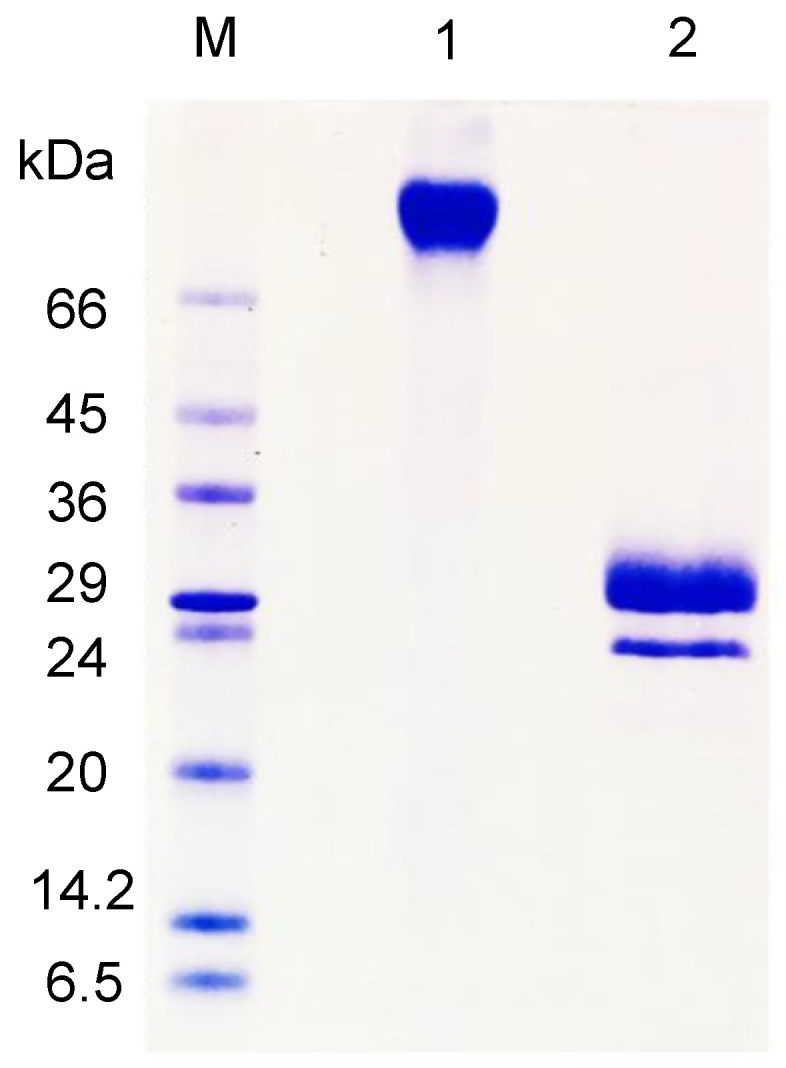
SDS–PAGE of purified AfiL. Lane M: molecular weight marker; Lane 1: AfiL under non-reducing conditions; Lane 2: AfiL under reducing conditions (with β-mercaptoethanol).

**Figure 2 microorganisms-13-01349-f002:**
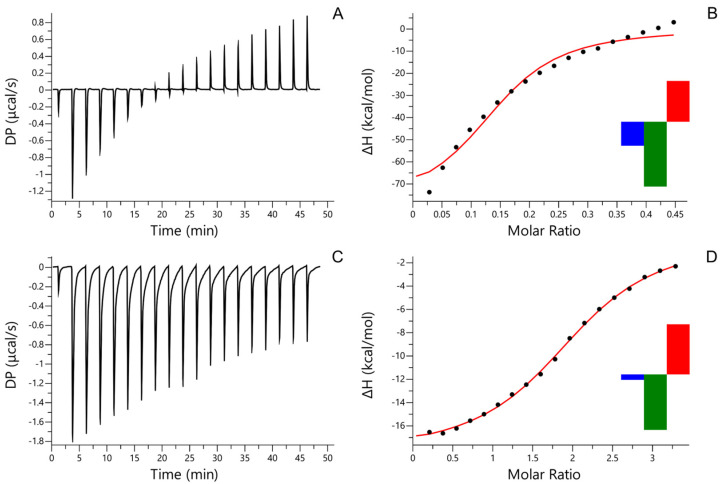
ITC. Isothermal titration calorimetry (ITC) of AfiL. (**A**) Thermogram of AfiL titrated with porcine stomach mucin (PSM). (**B**) Binding isotherm and model fitting for AfiL–PSM interaction; insert: heatmap showing the contributions of enthalpy (ΔH, green), entropy (−TΔS, red), and Gibbs free energy (ΔG, blue) to the binding AfiL–PSM interaction. (**C**) Thermogram of AfiL titrated with D-lactose. (**D**) Binding isotherm and model fitting for AfiL–lactose interaction; insert: heatmap representation of integrated heats for AfiL–lactose interaction. Experiments were performed at 25 °C using a MicroCal PEAQ-ITC system, and data were fitted to a one-set-of-sites binding model.

**Figure 3 microorganisms-13-01349-f003:**
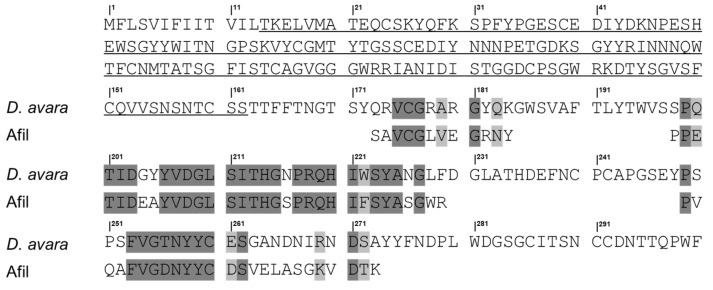
Sequence alignment between AfiL and a putative collectin-like protein from *Desydea avara*. Amino acid sequence alignment between AfiL (lectin from *Aplysina fistularis*), partially obtained by MS/MS, and a putative uncharacterized protein encoded in the genome of the marine sponge *Desydea avara* (GenBank ID: XP_065893106.1). Dark gray shading indicates identical residues; light gray shading indicates similar residues. Amino acids underlined correspond to a predicted fibrillar collagen-like domain.

**Figure 4 microorganisms-13-01349-f004:**
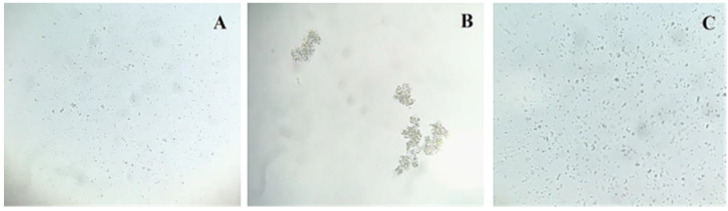
Effect of AfiL on bacterial cell agglutination. Microscopic visualization of bacterial cell agglutination induced by AfiL. (**A**) *S. aureus* in TBS (control); (**B**) *S. aureus* with AfiL; (**C**) *S. aureus* with AfiL pre-incubated with lactose. Images were acquired using light microscopy at 120× magnification. Agglutination was specifically observed for *S. aureus* and was inhibited in the presence of 100 mM lactose.

**Figure 5 microorganisms-13-01349-f005:**
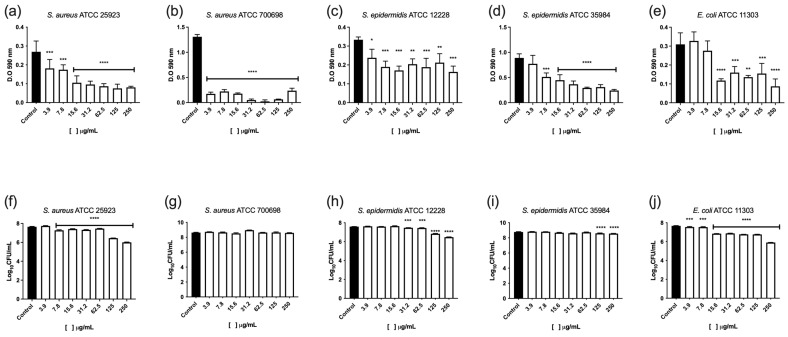
Effect of AfiL on biofilm formation of *S. aureus* and *E. coli* strains. Quantification of biofilm biomass (**a**–**e**) and viable cell counts (**f**–**j**). Bars represent AfiL-treated samples; black bars represent untreated groups (controls). Data are presented as mean ± standard deviation. Statistical significance was calculated in comparison to untreated controls using one-way ANOVA followed by the Bonferroni post hoc test: *p* < 0.05 (*), *p* < 0.01 (**), *p* < 0.001 (***), *p* < 0.0001 (****).

**Table 1 microorganisms-13-01349-t001:** Purification procedure of the lectin from *A. fistulares* (AfiL).

**Fraction**	**Total Protein** **(mg)**	**Titer** **(HU·mL^−1^)**	**Specific Activity** **(UH·mg^−1^)**	**Purification** **(x)**	**Yield** **(%)**
Crude Extract	67.5	1024	758.5	1	100
F 0–70%	70	1024	512	0.67	70
Affinity	1.2	512	6400	8.4	15

**Table 2 microorganisms-13-01349-t002:** Inhibition of the hemagglutinating activity of AfiL and ACL by sugars, glycoproteins, and glycoconjugates.

Carbohydrates	Minimum Inhibitory Concentration (MIC)
α-D-lactose	12.5 mM
D-lactose	12.5 mM
4-nitrophenyl-α-D-galactoside	12.5 mM
4-nitrophenyl-β-D-galactoside	12.5 mM
D-lactulose	25 mM
Glycoproteins	MIC
Porcine stomach mucin (PSM) type II	0.006 μg/mL
Porcine stomach mucin (PSM) type III	0.006 μg/mL
Bovine stomach mucin (BSM)	0.049 μg/mL
Fetuin	0.39 μg/mL
Bacterial Glycoconjugates	MIC
Lipopolysaccharide (LPS)	250 μg/mL
Lipoteichoic acid (LTA)	125 μg/mL

**Table 3 microorganisms-13-01349-t003:** Dynamic light scattering (DLS) parameters of AfiL under different pH conditions. Summary of hydrodynamic size data obtained for AfiL in acetate buffer (pH 5), phosphate buffer (pH 7), and glycine buffer (pH 10), including Z-Average diameter, polydispersity index (PDI), and main peaks identified in the intensity-based distributions.

pH	Z-Average	PDI	Peak 1 (nm) —Area	Peak 2 (nm) —Area	Peak 3 (nm) —Area
5	432.9	0.4171	223.6 (80.4%)	29.3 (19.6%)	-
7	475.7	0.4731	220.2 (62.9%)	45.4 (21.1%)	9.2 (16%)
10	168.5	0.5989	452.0 (45.9%)	118.4 (31.2%)	-

**Table 4 microorganisms-13-01349-t004:** Effect of AfiL combined with antibiotics on *S. aureus* and *E. coli* strains.

Bacterial Strains	Antibiotic			Effect	Antibiotic			Effect
	Ampicllin				Tetracycline			
	MIC ^a^ µg/mL	MIC ^b^ µg/mL	MIC ratio		MIC ^a^ µg/mL	MIC ^b^ µg/mL	MIC ratio	
*S. aureus* ATCC 700698	50	25	½	(AD)	100	12.5	1/8	(S)
*E. coli* ATCC 11303	0.39	1.56	2	(AN)	0.78	3.12	2	(AN)

MIC ^a^ = MIC value of antibiotic alone; MIC ^b^ = new MIC value of antibiotic combined AfiL; (S) Synergistic, (AD) Additive, (AN) Antagonistic.

## Data Availability

The original contributions presented in this study are included in the article and Appendix A. Further inquiries can be directed to the corresponding author.

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
