# Peer review of "Afil, a Lectin from Aplysina fistularis, Exhibits Antibiofilm and Synergistic Antibacterial Activity Against Resistant Bacteria"

_microorganisms, 2025, doi:10.3390/microorganisms13061349_

Round 1

Reviewer 1 Report

Comments and Suggestions for Authors

The authors present an interesting study on a novel lectin (AfiL) from Aplysina fistularis with promising antimicrobial properties. However, the manuscript requires substantial revisions. Specific concerns are outlined below:

(1) While the manuscript emphasizes AfiL's interaction with bacterial surface components (LTA/LPS), the rationale for focusing exclusively on Staphylococcus spp. and E. coli is unclear. Given the growing threat of multidrug-resistant Gram-negative pathogens, the limited scope weakens the claimed "broad-spectrum potential" of AfiL. Please clarify the selection criteria for bacterial strains.

(2) The claim that AfiL represents "the first functionally characterized collectin in sponges" is overstated. The MS/MS-derived sequence shows limited homology to a putative Dysidea avara protein, but no structural evidence is provided to support collectin classification.

(3) The antagonistic effect of AfiL-antibiotic combinations against E. coli (Table 4) contradicts the proposed "broad-spectrum" utility.

(4) The reconstructed 32-residue sequence (PPETLDEAYVDGLSLTHGSPRQHLFSYASGWR) includes multiple leucine (L) residues, but standard tandem mass spectrometry (MS/MS) cannot distinguish isoleucine (I) from leucine due to their identical molecular masses. The manuscript does not clarify how the authors resolved this ambiguity.

(5) The introduction lacks critical engagement with recent advancements in sponge lectin research (2024–2025)

(6) The discussion does not critically address the study’s limitations or outline actionable future directions 

(7) Font sizes and resolution in Figure5 are inadequate. Please improve.

(8) References such as 10-12 is incomplete (journal name missing). Standardize all references to comply with journal guidelines.

Comments on the Quality of English Language

The English could be improved to more clearly express the research.

Author Response

Dear Reviewer,

Please find below our point-by-point responses to the reviewers' comments. All revisions made in the manuscript are thoroughly addressed and explained accordingly.

#Reviewer 1

The authors present an interesting study on a novel lectin (AfiL) from Aplysina fistularis with promising antimicrobial properties. However, the manuscript requires substantial revisions. Specific concerns are outlined below:

Comment (1) While the manuscript emphasizes AfiL's interaction with bacterial surface components (LTA/LPS), the rationale for focusing exclusively on Staphylococcus spp. and E. coli is unclear. Given the growing threat of multidrug-resistant Gram-negative pathogens, the limited scope weakens the claimed "broad-spectrum potential" of AfiL. Please clarify the selection criteria for bacterial strains.
Response: We appreciate the reviewer’s insightful observation. We clarified in the Discussion that the bacterial strains were selected based on their contrasting cell wall architecture (Gram-positive vs. Gram-negative) and their clinical significance in biofilm-associated infections. We also acknowledged that the limited spectrum is a constraint and highlighted the need for future studies involving a broader range of resistant Gram-negative strains. Importantly, we revised the manuscript to remove or rephrase all mentions of “broad-spectrum potential” to more accurately reflect AfiL’s demonstrated strain-specific antibiofilm activity.

(2) The claim that AfiL represents "the first functionally characterized collectin in sponges" is overstated. The MS/MS-derived sequence shows limited homology to a putative Dysidea avara protein, but no structural evidence is provided to support collectin classification.
R: We thank the reviewer for this important observation. We agree that the original statement may have been overstated, particularly in the absence of complete sequence or structural confirmation. AfiL exhibited partial sequence similarity to a putative Dysidea avara protein, which in turn contains features reminiscent of collectins. However, as correctly noted, this alone is not sufficient to classify AfiL as a collectin.
Accordingly, we have revised the manuscript to clarify this point. Rather than suggesting that AfiL is a collectin, we now describe it as a "candidate for further investigation”, and explicitly recognize that the current evidence is preliminary and based on limited sequence homology. We believe this revision better reflects the data and avoids overinterpretation.
Changes were made in the Abstract and Discussion section (lines 537-538) to moderate the claim and align it with the evidence currently available.

(3) The antagonistic effect of AfiL-antibiotic combinations against E. coli (Table 4) contradicts the proposed "broad-spectrum" utility.
R: We revised the text to emphasize the strain-specific nature of AfiL's interactions and removed references to broad-spectrum action.

(4) The reconstructed 32-residue sequence (PPETLDEAYVDGLSLTHGSPRQHLFSYASGWR) includes multiple leucine (L) residues, but standard tandem mass spectrometry (MS/MS) cannot distinguish isoleucine (I) from leucine due to their identical molecular masses. The manuscript does not clarify how the authors resolved this ambiguity.
R: We thank the reviewer for highlighting this important point. Indeed, standard MS/MS techniques are unable to distinguish between leucine (L) and isoleucine (I) due to their isobaric nature. In our analysis, the assignment of leucine residues was based on enzymatic cleavage specificity. The peptides were generated through chymotrypsin digestion, and as chymotrypsin preferentially cleaves at the carboxyl side of aromatic residues and also at leucine residues (but not isoleucine), we inferred the identity as leucine when consistent with known cleavage sites.
We have now clarified this point in the Methods section (page 5, lines 204-208) to inform the reader that these assignments are based on enzyme-specific cleavage patterns.

(5) The introduction lacks critical engagement with recent advancements in sponge lectin research (2024–2025)
R: We thank the reviewer for this valuable suggestion. Initially, our introduction included a broader and more comprehensive review of the field, particularly highlighting studies from our research group that have contributed to the development of sponge lectin characterization. However, during the initial editorial screening, we were advised to reduce the number of self-citations to ensure a balanced and neutral tone, which led to the removal of several references.
Following the reviewer’s recommendation, we have now revised the introduction to incorporate relevant and recent studies from 2024 and 2025 by other authors. These include new insights into sponge lectin structure (now cited as references [3,4 and 9]). 

(6) The discussion does not critically address the study’s limitations or outline actionable future directions 
R: We agree. We added a new paragraph in the Discussion explicitly addressing study limitations and suggesting directions for future work.

(7) Font sizes and resolution in Figure5 are inadequate. Please improve.
R: We thank the reviewer for this observation. Figure 5 has been replaced with a high-resolution version. It was exported directly from GraphPad Prism at 600 dpi in TIFF format, with font sizes adjusted to ensure legibility in accordance with the journal’s standards. 

(8) References such as 10-12 is incomplete (journal name missing). Standardize all references to comply with journal guidelines.

We thank the reviewer for pointing out the inconsistency in our reference formatting. We have carefully reviewed and corrected all references. 

Reviewer 2 Report

Comments and Suggestions for Authors

This manuscript presents the purification and characterization of AfiL, a lectin from Aplysina fistularis, with a focus on its antibiofilm and synergistic antibacterial properties. The biochemical data are solid, and the overall experimental design is well executed. However, several biological claims require more cautious interpretation.

AfiL does not inhibit planktonic growth of most tested strains, with the exception of a methicillin-resistant S. aureus (MRSA) strain, where it exhibits a bacteriostatic effect at 125 µg/mL. Therefore, describing the compound as broadly “antimicrobial” is misleading. This observation, however, should not be viewed as a weakness of the study. It is common in antimicrobial testing that novel compounds show niche-specific or context-dependent activity. The fact that AfiL lacks planktonic activity does not diminish its scientific value. That said, the term “antibiofilm agent” is more appropriate and accurate than “antimicrobial” in this context.

One important clarification should be added to the discussion regarding the interpretation of viable cell counts within biofilms. I have worked with similar assays, but for readers unfamiliar with this type of experimental data, it may appear contradictory that biofilm biomass is drastically reduced while the viability of the remaining biofilm-associated cells remains high or even increases slightly. However, this is not a contradiction. The viable cell count reflects only the small population that still managed to establish a biofilm — and their high viability does not negate the compound’s strong antibiofilm effect. A brief explanation of this relationship would improve clarity and help avoid misinterpretation, especially by non-specialist readers.

Claims of therapeutic potential are premature in the absence of any cytotoxicity data. The authors should either include toxicity assays or explicitly discuss this limitation. Even a short paragraph contextualizing AfiL within the broader literature on sponge lectins — if they show any hemolytic or cytotoxic effects — would suffice.

Importantly, AfiL demonstrates robust and consistent antibiofilm activity against multiple clinically relevant strains, including S. aureus, S. epidermidis, and E. coli. In several cases, biofilm biomass was reduced by 80–90%, which is biologically significant. While the compound does not appear to be bactericidal, it clearly interferes with bacterial adhesion or early stages of biofilm development. This is particularly relevant for applications in preventing biofilm formation on medical devices or other abiotic surfaces. The synergistic interaction with tetracycline against MRSA is another noteworthy finding, although it is limited to a single strain and requires further validation.

Finally, the biochemical characterization of AfiL is a clear strength of the manuscript. The authors provide a comprehensive and well-executed suite of analyses — including SDS-PAGE, circular dichroism, DLS, ITC, and partial MS/MS sequencing — which give valuable structural insights and support the novelty of the compound.

In conclusion, the study is valuable within the field of biochemistry of lectins and anti-adhesion / anti-biofilm research. If the authors revise their conclusions to reflect the preventive rather than therapeutic role of AfiL, and temper their use of terms like “antimicrobial” or “therapeutic candidate,” the manuscript will be suitable for publication.

Author Response

Dear Reviewer,

Please find below our point-by-point responses to the reviewers' comments. All revisions made in the manuscript are thoroughly addressed and explained accordingly.

# Reviewer 2
Comment: This manuscript presents the purification and characterization of AfiL, a lectin from Aplysina fistularis, with a focus on its antibiofilm and synergistic antibacterial properties. The biochemical data are solid, and the overall experimental design is well executed. However, several biological claims require more cautious interpretation.
AfiL does not inhibit planktonic growth of most tested strains, with the exception of a methicillin-resistant S. aureus (MRSA) strain, where it exhibits a bacteriostatic effect at 125 µg/mL. Therefore, describing the compound as broadly “antimicrobial” is misleading. This observation, however, should not be viewed as a weakness of the study. It is common in antimicrobial testing that novel compounds show niche-specific or context-dependent activity. The fact that AfiL lacks planktonic activity does not diminish its scientific value. That said, the term “antibiofilm agent” is more appropriate and accurate than “antimicrobial” in this context.
Response: The designation of AfiL as an “antimicrobial” agent was reconsidered and revised throughout the manuscript. We now refer to AfiL primarily as an “antibiofilm agent” to better reflect its observed biological effects. Additionally, we removed or modified all instances implying broad-spectrum antimicrobial activity and clarified the compound’s selective or context-dependent activity in the manuscript. 

Comment: One important clarification should be added to the discussion regarding the interpretation of viable cell counts within biofilms. I have worked with similar assays, but for readers unfamiliar with this type of experimental data, it may appear contradictory that biofilm biomass is drastically reduced while the viability of the remaining biofilm-associated cells remains high or even increases slightly. However, this is not a contradiction. The viable cell count reflects only the small population that still managed to establish a biofilm — and their high viability does not negate the compound’s strong antibiofilm effect. A brief explanation of this relationship would improve clarity and help avoid misinterpretation, especially by non-specialist readers.
R: We thank the reviewer for this important suggestion. We fully agree that, without clarification, the observed discrepancy between reduced biofilm biomass and relatively high viable cell counts may appear contradictory to non-specialist readers. Accordingly, we have added a brief explanation to the Discussion, as suggested.

Comment: Claims of therapeutic potential are premature in the absence of any cytotoxicity data. The authors should either include toxicity assays or explicitly discuss this limitation. Even a short paragraph contextualizing AfiL within the broader literature on sponge lectins — if they show any hemolytic or cytotoxic effects — would suffice.
R: Response: We appreciate the reviewer’s important observation. While we did not perform formal cytotoxicity assays in this study, the hemagglutination assays conducted with AfiL support the notion that the lectin does not exhibit hemolytic activity, as no hemolysis was observed in the tested conditions.

Comment: Importantly, AfiL demonstrates robust and consistent antibiofilm activity against multiple clinically relevant strains, including S. aureus, S. epidermidis, and E. coli. In several cases, biofilm biomass was reduced by 80–90%, which is biologically significant. While the compound does not appear to be bactericidal, it clearly interferes with bacterial adhesion or early stages of biofilm development. This is particularly relevant for applications in preventing biofilm formation on medical devices or other abiotic surfaces. The synergistic interaction with tetracycline against MRSA is another noteworthy finding, although it is limited to a single strain and requires further validation.
R: We thank the reviewer for recognizing the strength and consistency of the antibiofilm activity observed with AfiL. We agree that these results are biologically relevant, particularly in the context of preventing biofilm formation on medical devices and abiotic surfaces. In response to the reviewer’s comment, we also acknowledge in the revised manuscript that the observed synergistic effect with tetracycline was limited to a single MRSA strain, and that further validation with a broader range of clinical isolates will be essential in future studies. This limitation is now explicitly discussed in the revised Discussion.

Comment: Finally, the biochemical characterization of AfiL is a clear strength of the manuscript. The authors provide a comprehensive and well-executed suite of analyses — including SDS-PAGE, circular dichroism, DLS, ITC, and partial MS/MS sequencing — which give valuable structural insights and support the novelty of the compound.
R: We thank the reviewer for recognizing the strength of our manuscript.

Comment: In conclusion, the study is valuable within the field of biochemistry of lectins and anti-adhesion / anti-biofilm research. If the authors revise their conclusions to reflect the preventive rather than therapeutic role of AfiL, and temper their use of terms like “antimicrobial” or “therapeutic candidate,” the manuscript will be suitable for publication.

Round 2

Reviewer 1 Report

Comments and Suggestions for Authors

The authors have addressed my previous concerns. However, please ensure that the formatting of references 22 and 33 is corrected according to the journal's style guide.